# Remarks on Selected Morphological Aspects of Cancer Neuroscience: A Microscopic Photo Review

**DOI:** 10.3390/biomedicines12102335

**Published:** 2024-10-14

**Authors:** Ewa Iżycka-Świeszewska, Jacek Gulczyński, Aleksandra Sejda, Joanna Kitlińska, Susana Galli, Wojciech Rogowski, Dawid Sigorski

**Affiliations:** 1Department of Pathology and Neuropathology, Medical University of Gdansk, 80-210 Gdansk, Poland; ewa.izycka-swieszewska@gumed.edu.pl; 2Department of Pathomorphology, Copernicus Hospital, 80-803 Gdansk, Poland; 3Department of Pathomorphology an Forensic Medicine, Collegium Medicum, University of Warmia and Mazury, 10-561 Olsztyn, Poland; 4Department of Biochemistry and Molecular and Cellular Biology, Georgetown University Medical Center, Washington, DC 20057, USA; jbk4@georgetown.edu (J.K.); sbg46@georgetown.edu (S.G.); 5Institute of Health Sciences, Pomeranian University, 70-204 Slupsk, Poland; 6Department of Oncology, Collegium Medicum, University of Warmia and Mazury, 10-228 Olsztyn, Poland

**Keywords:** cancer neuroscience, perineural invasion, nerve density, neurotransmitters, nerve pathology, neural factors, axonogenesis

## Abstract

Background: This short review and pictorial essay presents a morphological insight into cancer neuroscience, which is a complex and dynamic area of the pathobiology of tumors. Methods: We discuss the different methods and issues connected with structural research on tumor innervation, interactions between neoplastic cells and the nervous system, and dysregulated neural influence on cancer phenotypes. Results: Perineural invasion (PNI), the most-visible cancer–nerve relation, is briefly presented, focusing on its pathophysiology and structural diversity as well as its clinical significance. The morphological approach to cancer neurobiology further includes the analysis of neural density/axonogenesis, neural network topographic distribution, and composition of fiber types and size. Next, the diverse range of neurotransmitters and neuropeptides and the neuroendocrine differentiation of cancer cells are reviewed. Another morphological area of cancer neuroscience is spatial or quantitative neural-related marker expression analysis through different detection, description, and visualization methods, also on experimental animal or cellular models. Conclusions: Morphological studies with systematic methodologies provide a necessary insight into the structure and function of the multifaceted tumor neural microenvironment and in context of possible new therapeutic neural-based oncological solutions.

## 1. Tumor Neural Microenvironment: General View and State of the Art

The relations between the nervous system and cancer has been under investigation for a long time, and these investigations have accelerated significantly in recent years. Tumor neurobiology has evolved into a rapidly developing scientific area with complex methodologies. Nervous system regulation and neural signaling inherently contribute to embryogenesis, tissue remodeling, homeostasis, plasticity, and regeneration. The close relations among fetal cells, blood vessels, and nerves are important in normal development and body formation. During the reconstitution of lost body parts in some animals, nerve ingrowth and axonogenesis accompany angiogenesis and inflammation within the transforming blastema. Not unexpectedly, axono-/neurogenesis processes and nerve reprogramming have also been documented in many neoplasms [1,2]. Reciprocal, aberrantly activated interactions between cancer cells and all levels of the nervous system, as well as the neural component of the tumoral microenvironment (TME), significantly affect multiple hallmarks of cancer. In parallel, growing tumors induce neural remodeling, support nerve ingrowth neuronal excitability, and reinforce cancer–nerve interactions [3,4,5,6]. Structural research includes various types of morphological–histological assessment of human oncological samples, animal models, tissue cultures, or organoids, as well as different types of radiological imaging. Many aspects and challenges of morphological analyses of neural TMEs exist, which can be explored in the field of neural transmission, electrochemical, autocrine, and paracrine mechanisms, systemic nervous system–cancer interactions, and oncological therapy’s influence on the nervous system [7]. Functional studies of cancer neurobiology use electrophysiological analyses, voltage and calcium imaging, optogenetics, intravital imaging, modifications and measurements of nerve signaling, surgical denervation models, and stress studies. Molecular methods are represented in a diverse range of genomic and expression studies as well as single-cell/spatial multiomics [8,9].

In 2007, a hypothesis of neoneurogenesis and neuro-neoplastic synapsis was raised [10]. Ayala’s group was one of the first who described direct cancer–nerve interactions, axonogenesis, and neurogenesis within the dorsal ganglia in prostate cancer (PCa) [11,12,13,14]. Further studies have shown the involvement of the autonomic nervous system in cancerogenesis and that nerves regulate early- and late-PCa tumor growth, while denervation suppresses tumorigenesis and metastases in animal models [15,16,17]. It was shown that adrenergic signaling from sympathetic nerves recruited into the tumor promotes prostate cancer growth via beta-receptors on stromal cells and that peripheral malignancies attract neural progenitor cells from the brain to the growing tumor mass [18]. It has become clear that the role of nerves and neural signaling in neoplasia is multidirectional. The nerves interfere with the tumor angiogenesis process, together with the modulation of nutrients, oxygen, and growth signal supply. In addition, neural signals sustain tumor proliferation and induce resistance to apoptosis, stimulate cancer stem cell compartments, and lead to tumor-related neuropathic pain. The next important finding concerns CNS/PNS/autonomic mediation of immune and inflammatory responses to neoplasia [19]. Through influencing cellular homeostasis and energetic metabolism, the nervous system modifies metabolic adaptation and reprogramming [12,20]. It has been proposed in a simplified idea that axonogenesis is the first phenomenon that supplies tumor growth, and neuronal transdifferentiation can be the last form of cancer cell independence in some malignancies [15,17,21]. Sympathetic and parasympathetic signaling involves many processes but has different effects and engagement with various tumors [22]. For instance, parasympathetic nerves in gastrointestinal cancer stimulate proliferation through cholinergic signaling, since PCa models connect them with metastatic dissemination [15,23]. Additionally, the contribution of sensory nerves was encountered in breast, pancreatic, skin, and head and neck cancers, mainly in immune response modulation. The sensory nerves were considered as drivers of bone metastases [24,25,26].

Communication between tumor stroma and the peripheral nervous system occurs through neurotransmitters, axon guidance molecules, and neurotrophic factors [12,16]. The key autonomic neurotransmitters are noradrenaline (ADR/β receptor) and acetylcholine (rec M) in the sympathetic and parasympathetic systems, respectively. Noradrenaline released from nerves upregulates angiogenesis (↑VPF, VEGF, and MMPs), leading to a metabolic switch. ADR signaling affects cytokines and immunity (↑TGF-β, ↓pro-apoptotic cytokine and NK cells), tumor-associated macrophages (macrophage M2 polarization), tumor microenvironment status (↑CAF, proinflammatory cytokines, and modulation macrophage M2), oncogenic pathway status (ARRB1, PKA, and oncogene Src activation, and hTERT expression) contributing to cancer progression [22,27]. Other identified pathways include glutamate-mediated signaling via NMDA receptors, GABA, and FMRP regulation, including immunosuppression in TMEs. In addition, several prosurvival pathways, such WNT or Hedgehog can be activated through neurosignaling [23]. Another group of neurotransmitters with auto- and paracrine functions, detected in cancer stroma, constitutes neuropeptides, such as neuropeptide Y (NPY), substance P, neurotensin, orexin/hypocretin, somatostatin, and vasoactive intestinal peptides [28]. NPY secreted by the nerves can be also expressed by tumor cells in relation to their molecular characteristics and even the topography of the infiltration [29,30].

Many substances secreted by nerves, inter alia, promote chemotaxis, tumor cell invasion, and in parallel promote the growth and extension of axons toward cancer cells. The reciprocal interactions create loops of dependences affecting/reprogramming the molecular mechanisms in cancer cells, neural elements, and TME components such as stromal fibroblasts, Schwann cells, and immune system cells [31,32]. Perineural invasion represents another direct form of cancer–nerve crosstalk. However, the neoplastic cells not only undergo reprogramming, since in head and neck malignancies, since the transdifferentiation of tumor-associated sensory nerves into the adrenergic phenotype induced by cancer-derived extracellular vesicles has been described [26].

The important neurobiological players within many tumors are neuroendocrine cells with paracrine and autocrine functions due to the release of hormones, neuropeptides, and growth factors. The neuroendocrine cell population and level of such differentiation in tumors can present a wide spectrum, depending on the disease stage and phase of therapy [33]. The development of neuroendocrine differentiation (NED) is one of the mechanisms of cancer progression in the late stages of hormone-resistant PCa [34]. Transdifferentiation toward an aggressive small-cell neuroendocrine cancer (SCNC) phenotype occurs in response to targeted therapies, for example, in resistance to epidermal growth factor receptor (EGFR) kinase inhibitors in lung adenocarcinomas [35].

It seems that cancer–nerve crosstalk and the neural TME composition depend on tumor type, stage, molecular profile, the involved organs, and numerous unrecognized factors. Tumor grade, pattern of neoplastic invasion (solid, dispersed, or tumor budding), and cancer stroma features have an impact too. It is probable that tumor innervation features are partially unique for each case. Some tumors are clearly nerve-dependent with activation of axonogenesis and cellular nerve affinity. In some types of neoplasia, the innervation is significantly lower than in the host tissue; thus, perhaps dysregulation/deprivation and downregulation of neural control participate in tumorigenesis there [36]. A morphological analysis of the tumor microenvironment has shown various distributions of nerves and diverse so-called “neuroepithelial interactions” [21]. These molecular and functional contributions remain a challenging issue for exploration for neuroscientists and pathologists.

## 2. Pathological Aspects of Neural TME Investigation

### Basic Methods and Tissue Markers

The techniques applied in the morphological analyses of cancer neuroscience issues comprise light microscopy with immunohistochemistry (IHC), histochemistry, and immunofluorescence techniques. Deeper insights can support electron microscopy, confocal microscopy, as well as recent spatial transcriptomic analyses. Image digitalization, virtual slides, and artificial intelligence are useful tools in morphological investigations. In vitro studies usually rely on measuring changes in neuronal morphology, including neurite length, branches, or number, but modern techniques such as gene-editing-related methods like CRISPR/Cas9 or gene expression analysis can also be used [16,37,38].

Bigger nerves and ganglia within and around tumor are evident in routine HE staining, but smaller nerve branches and single axons are invisible (Figure 1). Thus, many immunohistochemical markers for cytoskeleton elements, structural and secretory proteins, various receptors, neurotropic factors, enzymes, and signal transducers are suitable [5,16]. Non-selective nerve fiber markers include PGP 9.5, S100, CD56, neurofilaments, L1CAM, synaptophysin, and calretinin (Figure 2). Selective markers enable the identification of sympathetic (anti-tyrosine hydroxylase (TH)), parasympathetic (anti-vesicular acetylocholine esterase (VACHT)), or sensory fibers (CGRP, substance P). Dendrite markers include MAP2, SAP102, Debrin 1, ARC, GAP-43, Neuropiline-1, RNT-1, and SNAP25. Antibodies against NF200 are used for large-diameter axon delineation, and beta 3 tubuline and PGP9.5 are used for small-fiber/-axon detection [5,30,39]. Other markers include NeuroD1, NeuN, Microtubule-associated protein 2 (MAP2), chromogranin, as well as more selective profiled detectors of specific nerve subtypes. In addition, levels of neural differentiation and maturation can be assessed with SOX2, SOX9, and Netrin-1. For Schwann cells, antigen S100 is suitable, while EMA, CD34, and Glut1 are expressed by perineurial cells. There are also many good histochemical methods that have been developed a long time ago, such as Kluver–Barrera stain, toluidine blue stain, silver impregnation methods, safranian stains, Masson’s trichrome stain, etc. [5,40]. In summary, the most frequently used IHC markers for tumor innervation studies are S100, PGP9.5, NFP, synaptophysin, NeuN, calretinin, and TH in classical and fluorescent microscopy.

Neuroendocrine cells and differentiation occurring in a wide range of cancer types [41,42,43,44] can be morphologically evaluated with routine markers such as synaptophysin, chromogranin A, CD56, and neuron-specific enolase. Somatostatin receptors, diverse neurotransmitters, and neuropeptides (VIP, NPY, galanin, substance P, and neurotensin) are also examined in this context. The mechanisms contributing to NE switching in cancer are diverse but not well understood and not yet discovered [45].

Morphological–neuropathological investigations of tumors comprise the assessment of neural networks connected with the tumors, distribution, types of nerve, neural density, and complex aspects of perineural invasion.

## 3. Structural Assessment of Neural Networks (Topography) within Tumors

Regularly distributed neural networks present in normal organisms and organs are derived from physiological–embryonal neurogenesis, precise axonal extension during the development of organ/tissue innervation, as well as neural regeneration [1]. In the microenvironment of growing tumors, neural networks are dynamically transformed and usually irregular. The pre-existing nerves innervate the host organ’s cellular–structural elements and the blood vessels, since the other small nerve endings have their origin mainly from axonal sprouting [46]. The altered innervation in cancer results from neoplastic infiltration and displacements within the host tissue and its surroundings, as well as axonogenesis and central/peripheral neurogenesis [14,18]. Cancer cells and tumoral stroma are suspected to initiate nerves to sprout toward the tumor and support its growth from the early stages. Interestingly, the neurite-like protrusions and tumor microtubes in brain tumors allow neoplastic invasion, glioma proliferation, and support network function [47]. Likewise, axon-like cellular processes have been recently described as supporting the migration and metastatic potential of small-cell lung cancer [48]. This raises the question about the existence and role of synapses or synapse-like structures between neurons and cancer cells [49,50]. (Figure 3). It seems that, similarly to tissue regeneration, different types of nerves are responsible for angiogenesis, migration, and cellular division. There are three main sources of cancer innervation: axons already present in the transformed tissue, new fibers arising from nerves localized in the vicinity of tumor tissue (axogenesis), and neurons migrating/induced from the distal part of the central and peripheral nervous system [2,6,10]. Recent data suggest that cancer stem cells, as well as some populations of macrophages, can transform into neural progenitors within TMEs [51]. The invasion front of the cancer tissue interacts with pre-existing nerves in many forms, including, among others, destruction, incorporation, or growth stimulation. Structural studies on tumor innervation maps including all levels from bigger branches to small fibers on whole sections would give a comprehensive view (Figure 4).

Building upon knowledge about the morphology of tumoral neural network distribution in biologically different areas in quantitative and qualitative contexts is necessary. A promising tool that allows for deeper insight into tissue architecture, cellular heterogeneity, processes of tumorigenesis, and cancer spread is three-dimensional (3D) visualization reconstruction [40,52,53].

## 4. Perineural Invasion in Neoplasms

The major routes of cancer spreading consist of direct invasion into surrounding tissues, lymphatic and hematogenous metastasis, seeding along body cavities, as well as perineural invasion (PNI) [21,31,54]. The ability to invade the neural structures is common in head and neck, prostate, and digestive system (pancreatic, biliary, gastric, and colorectal) cancer, but its role and significance is still unclear. In many tumors, the presence, extent, and type of PNI is a as prognostic factor correlating with the risk of relapse, survival, and metastases [21,31]. The spread of perineural tumors into large nerve branches may be detected radiologically, but histopathological examinations remain as the optimal PNI assessment of smaller nerve branches (Figure 5).

PNI, being the pathway of continuous invasion and having input in metastatic spread (postulated bone involvement), constitutes and creates the specific milieu for cancer cells, called the perineural niche. Normally, this niche regulates the physiology of nerves and their development, growth, and regeneration. It is a unique microenvironment background for PNI, composed of Schwann cells, blood vessels, fibroblasts, extracellular matrices, and immune cells [31,32,37]. Schwann cells play a particular role in these spaces sustaining the repair, development, and trophic support of nerves as well as promotion of its spread through the recruitment of neoplastic cells, cell protrusion formation, motility support, metabolic switching, and interactions with the immune system [55,56,57]. Schwann cells, through the release of chemokine (C-C motif) ligand 2, attract M2 macrophages, which are responsible for tumor cells’ survival signaling, local immunology, and therapy resistance [56,58]. The tumor-associated nerves also regulate immune response and activity through neurotransmitters and immune checkpoints [19]. There is a variety of factors involved in the regulation of tumor–nerve interactions; among others, these are neurotransmitters and their receptors (NE/ADR and Ach/AChR), neurotrophins and their receptors (NGF/TRK/p75NTR, NT-3/TRKC, NT-4/5, BDNF/TRKB, and GDNF/RET/GFRα), chemokines and their receptors (CX3CR1/CX3CL1, CCR2/CCL2, and CXCR4/CXCL12), metalloproteinases or cell-surface molecules and their receptors (L1-CAM, NCAM, semaphoring 3A/plexins A1-A4, neuropilin-1, and semaphorin 4F) [57,59]. Moreover, the extracellular vesicles bridge some events in PNI crosstalk such as exosomes, which can contain and release neurotransmitters and factors regulating neurite growth and axonal regeneration [60]. For example, NPY—a pleiotropic factor acting through its receptors YR1-5—is present mainly in adrenergic nerve fibers but also in exosomes [29,61].

Morphologically, nerve branch infiltration might be found in HE sections (Figure 6) and visualized in more detail when supported by ancillary methods such as histochemical and immunohistochemical staining [62,63]. PNI status is incorporated into routine diagnostic synoptic pathological cancer reports in dual form: present or absent, and if present, it can be described as focal or multifocal/diffuse. The criteria are, however, not clearly defined. The most commonly used PNI definition refers to neoplastic infiltration of any nerve layer, epineurium, endoneurium, or perineurium; however, the last of these is the most commonly affected [31,54]. The patterns of invasion include intraneural encirclement, complete and incomplete (“crescent-like”) encirclement, partial invasion, concentric “onion skin” lamination, and neural permeation [64]. Because of difficulties in the standardization of definitions and methods of visualization of PNI and to increase the level of its detection, deep learning techniques combined with artificial intelligence and bioinformatic analysis are also being developed. Growing cancer masses cross the anatomic neural structures, for instance, enteric plexuses, and infiltrate nerve branches of different sizes, which can lead to the disintegration of nerves into separate fibers/axons within the tumor stroma (Figure 7). The creation of basket-/net-like small-fiber structures around malignant cells creating new forms of direct contact has been observed [62]. In the majority of studies, PNI in tumors is quantified dichotomously into PNI-positive (when at least one nerve is affected) and negative cases [65,66,67]. However, some authors propose additional parameters for quantitative PNI description. Miller et al. suggest criteria for distinguishing between extratumoral and intratumoral locations when the distance from the tumor edge is less than 0.2 mm [66,68]. In other studies, the number of infiltrated nerve fibers per high-power field was evaluated and found to be informative [69]. Qualitative parameters usually comprise complete or incomplete nerve encirclement [70].

PNI represents the interaction between cancer cells and nerves driven by signaling molecules, having a complex impact on malignant cells [31,57] (Figure 8). It influences axonal transport and enables the remodeling of the network transmission in the upper parts of PNS/CNS. These interactions may be evaluated in vitro in cell co-cultures, migration assays, and tissue explants or in vivo in implantable tumors in animal model experiments. (Figure 9). Implementing multiplexed imaging methods (PET, CT, and MRI) together with multimodal genomic, transcriptomic, proteomic, and epigenomic analyses supports a considerable amount of data. The development of the spatial reconstruction models of tumoral innervation for functional PNI research is necessary for answering many questions [71].

## 5. Axonogenesis and Tumor Neural/Axonal Density

One of the most commonly used parameters for describing tumor innervation is nerve density (ND), usually defined as a number of nerve profiles within a particular field of view; however, there is no standardized definition [15,36,62]. Other studies use the idea of neural area as the area positive for nerves divided by the total counted area. There are various assessment methods and parameters used for ND description, which lead to variegate quantitative results and discrepancies between the studies. In the literature, nerve/axon density values have wide range; e.g., in prostate cancer, it is reported quantitatively, from less than 5 to about 100 per field, or just as “low” vs. “high” depending on the methodological approach [15,62,72,73]. The differences in approach include the type of specimen (tissue microarrays, tumor sections, or whole tumor sections), the size of the observation field, and the size of the counted nerves from nerve branches to single axons (Figure 10). Important questions relate to the topographic aspects of assessment (central versus peripheral tumor areas), tissue section thickness, the ancillary studies used for nerve detection, the methodology used for nerve counting, and manual versus automated or artificial intelligence involvement [36,62,74,75,76,77,78,79,80,81,82,83]. Higher ND is thought to be connected with increased proliferation and aggressiveness of malignancies. However, these data are inconsistent for different tumor types because of methodological discrepancies. Patho-clinical correlations have shown higher nerve density to be connected with worse prognoses and higher rates of metastasis and recurrences. In parallel, other studies have disclosed significant correlations between lower ND and poor outcomes [14,15,23,74,75,76,77,78].

ND was investigated in different tumor types, but the most widely researched is prostatic carcinoma. Initial observations have shown higher nerve density inside prostatic cancer and peritumoral areas than in normal prostate tissue and have shown cancer cells interacting with neurons to promote neurite growth [11,79]. However, the distribution of the axonal network in PCa differs between the invasion front of the tumor and peripheral parts of cancer infiltration. The sympathetic nerve fibers are located in the cancer vicinity, contrary to parasympathetic fibers situated more frequently within the central tumor area. The highest concentration of nerves and PNI (multifocal or unifocal) is observed in the prostate peripheral subcapsular region. It is important to stress that during PNI, well-defined nerve bundles change into individual fibers within cancer infiltrate, so many axons are pre-existing remodeled neural elements [15,62]. Also, in thyroid and pancreatic cancer, most of the nerves occur predominantly in the peripheral regions of the tumor or its surroundings [77,80]. Many studies disclose lower ND within pancreatic-cancer-central parts than in normal pancreas [78,81]. In colorectal cancer, sympathetic fibers were observed in the stroma adjacent to cancer cells, whereas parasympathetic fibers in the stroma were further away from cancer cells [82]. In breast cancer, nerves are more numerous in the invasion front of the tumor than in their central part [63]. Interestingly, PGP9.5-positive nerve fibers were not seen in high concentrations within the stroma of ductal carcinoma in in situ and benign fibroadenoma [84]. However, distinct neural fiber populations may act differently in relation to tissue type, histogenesis, and tumor molecular type. There are limited data on the molecular mechanisms driving tumor innervation. In oral squamous cell carcinoma, the loss of the p53 protein drives tumor-associated nerve reprogramming. The vesicles released by tumor cells drive neuronal adrenergic switching by losing miR-34, and ND increases and is correlated with worse outcomes [26]. It has been postulated that cancerous tissue contributes to changes in the morphology and plasticity of nerves. The enteric plexus within colon tumors can be reduced in size without evidence of nerve fibers. Cancer invasion destroys the local enteric nervous system and induces subsequent atrophy of the submucosal and myenteric plexuses in areas adjacent to the cancer boundary. Interestingly, it is accompanied by the increased number of galanin-immunoreactive neurons and increased neuroprotective galanin content in parts of the intestine close to the tumor [84]. Neural network remodeling also occurs in pancreatic cancer and includes hypertrophy of the nerves with increased nerve fiber density [85]. Moreover, even hematological malignancies interact with nerves within the bone marrow or spleen. Multiple myeloma recruits and activates the nerves from the bone microenvironment, leading to increased periosteal innervation and the infiltration of neuronal structures, additionally causing bone pain [86].

## 6. Neuroendocrine Differentiation in Neoplasms

Neuroendocrine cells, as a source of neurotransmitters and neuropeptides, present in many types of cancer, share its activity with the nerves and neoplastic cells in auto- and paracrine manner [43,44]. The heterogeneous morphological pattern of NED in tumors may be focal, having isolated cells, cell clusters, or dispersed or diffuse arrangements. Multiple approaches to NED assessment exist, depending on the examination method, evaluation criteria, and panel of possible compatible markers (Figure 11). The most commonly used assessment methods determine a specific quantity of cells per the defined examined field or percentage of immunopositive cells per field or area, as well as different indexes and cut-off levels [33,34,62]. However, the topographic assessment of neuroendocrine markers or neuropeptides also seems to be relevant. For instance, detailed NPY expression analyses in some types of neoplasia showed increased expression in PNI areas and invasion fronts of prostate and pancreatic cancer (Figure 12). This suggests NPY’s involvement in perineural trafficking and shows topographically dependent phenotypic features of heterogenous tumor masses [29,30,73].

## 7. Final Remarks

The neural regulation of malignancy appears as a multidimensional phenomenon and has been introduced as a new hallmark of cancer [3,72]. Modern systematic morphological and morphometric analyses of widely characterized tumor neural microenvironments at the functional/molecular level are necessary for dynamic advancement in cancer neuroscience and neural-based therapies.

The idea of our study was to present a photo report with a short review (some kind of pictorial essay). Morphological images are not often and widely presented in the current literature; moreover, we believe that simple but detailed visualization of different cancer tissue sections explains many aspects of cancer–nerve crosstalk and can inspire scientists to create new methodological approaches and solutions. The current challenges in cancer neuroscience in relation to morphological aspects include understanding the complex, bidirectional communication networks between cancer and nerves that contribute to cancer progression, metastasis, and resistance to therapy. Future directions in cancer neuroscience include advanced imaging techniques; molecular profiling; AI-driven digital pathology; deciphering molecular drivers and specific biomarkers; and translating findings from the lab to clinical practice, which may lead to new therapeutic strategies.

## Figures and Tables

**Figure 1 biomedicines-12-02335-f001:**
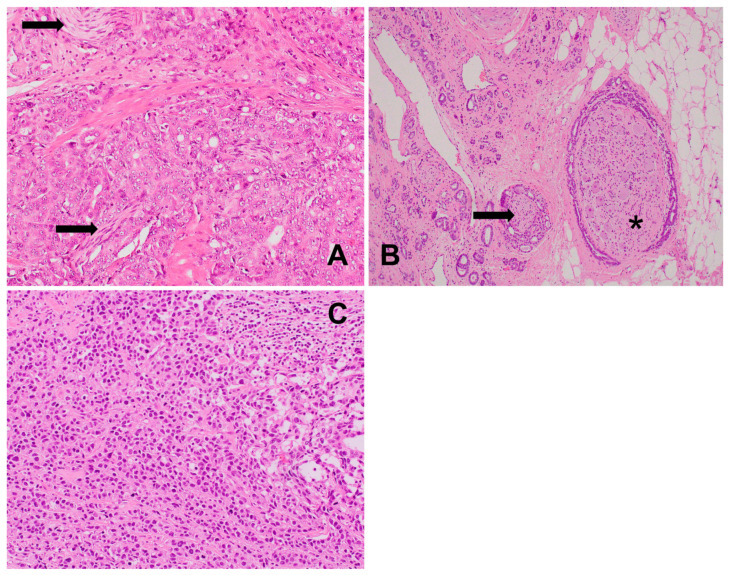
(**A**) Poorly differentiated prostatic adenocarcinoma: only bigger nerves (arrows) embedded within the neoplastic tissue are visible, HE, 200×. (**B**) Well-differentiated prostatic adenocarcinoma: perineural invasion with glandular structures cuffing the ganglion (*) and nerve branches (arrow) at the periphery and extraprostatic tumor extension, HE, 100×. (**C**) Poorly differentiated prostatic adenocarcinoma: solid growth pattern with mild inflammatory infiltrate (upper-right corner); no nerve fibers are visible at all; HE, 200×.

**Figure 2 biomedicines-12-02335-f002:**
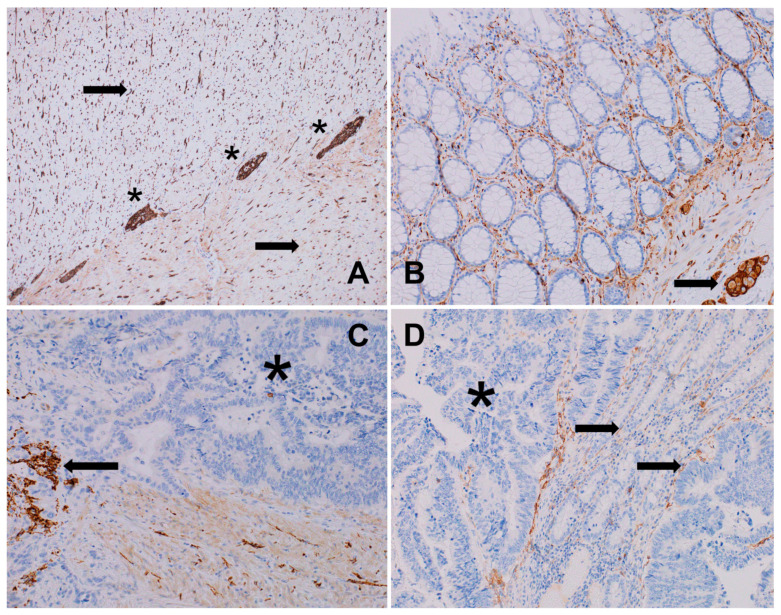
(**A**) Normal intestinal wall with a regular, dense network of axons within the muscular layers of the colon (arrows) and Auerbach’s myenteric plexus (*), S100, 100×. (**B**) Normal intestinal mucosa and submucosa presenting a regular, small-fiber network and Meissner plexus (arrow), CD56, 200×. (**C**) Colon adenocarcinoma: disruption and paucity of the axonal framework inside the cancer infiltrate (*) with the preservation of condensed or pre-existing axons (arrow) at the stroma outside the infiltrate, S100, 200×. (**D**) Well-differentiated colon adenocarcinoma: interface between normal intestinal mucosa and carcinoma in a normal gut; regular innervation is seen (arrows); only single axons are present in cancer infiltrate (*); CD56, 200×.

**Figure 3 biomedicines-12-02335-f003:**
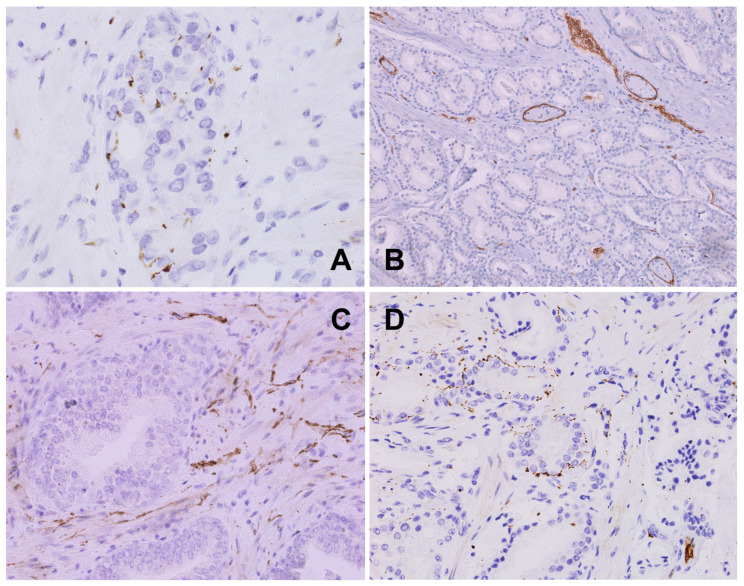
(**A**) Prostatic adenocarcinoma: multiple small fibers intermixed with neoplastic cells forming new cancer–nerve interactions, PGP 9.5, 600×. (**B**) Prostatic adenocarcinoma: sheath nerve branches embedded within cancer infiltrate visualized with perineural marker GLUT1; intravascular red blood cells are also stained; GLUT1, 200×. (**C**) Hyperplastic prostate: dense, slightly irregular axonal network within the periglandular stroma in the periphery of cancer (TH, 400×). (**D**) Prostatic adenocarcinoma with many axons among the infiltrating neoplastic glands, PGP9.5, 200×.

**Figure 4 biomedicines-12-02335-f004:**
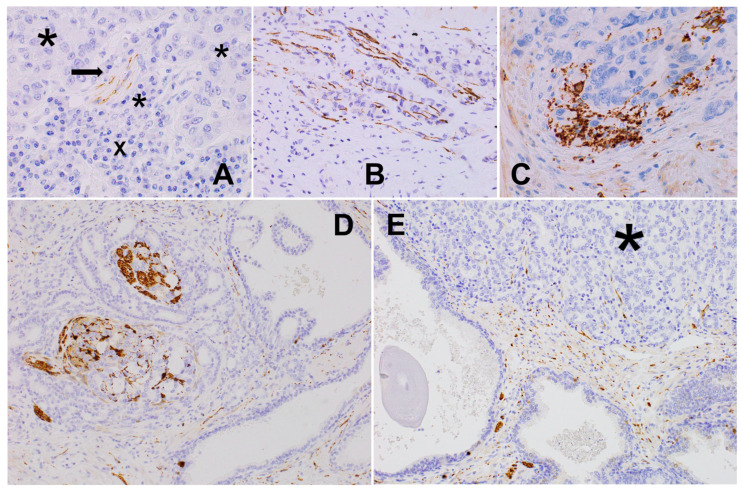
(**A**) Pancreatic adenocarcinoma: small nerve branch (arrow) almost completely destroyed by cancer cells (*) at the tumor invasion front accompanied with marked lympho-plasmocytic infiltrate (X), TH, 600×. (**B**) Poorly differentiated pancreatic adenocarcinoma: axonal network among tumor cells at the invasion front, probably based on destroyed, dissected, pre-existent nerves, PGP 9.5, 600×. (**C**) Pancreatic adenocarcinoma: accumulation of axons at the front of neoplastic infiltrate, destructed ganglion can be seen, PGP 9.5, 600×. (**D**) Prostate adenocarcinoma (PCa): peri- and intraneural invasion and multiple single axons within the surroundings, PGP 9.5, 200×. (**E**) Axonal distribution in prostate cancer (*) at the invasion front; sprouting from pre-existing prostatic stroma into high-grade infiltrate PGP9.5, 200×.

**Figure 5 biomedicines-12-02335-f005:**
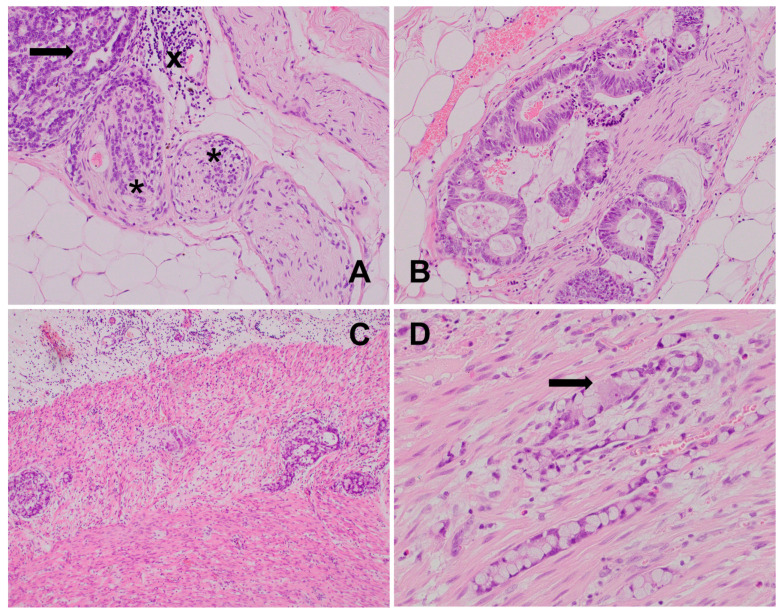
(**A**) Breast ductal adenocarcinoma (arrow) with the intraneural invasion (*) accompanied by a chronic perineural and focally intraneural inflammatory lymphocytic infiltrate (X), HE, 200×. (**B**) Colorectal adenocarcinoma stage pT3-diffuse invasion of cancer glands with mucin deposits inside the nerve branch within the mesenteric adipose tissue, HE, 200×. (**C**) Appendiceal goblet cell carcinoma: mucin-producing neoplastic cells spreading along the Auerbach’s plexus, HE, 100×. (**D**) Appendiceal goblet cell carcinoma with individual neoplastic cells directly contacting ganglion cells inside the myenteric plexus (arrow): physical cancer–nerve crosstalk, HE, 400×.

**Figure 6 biomedicines-12-02335-f006:**
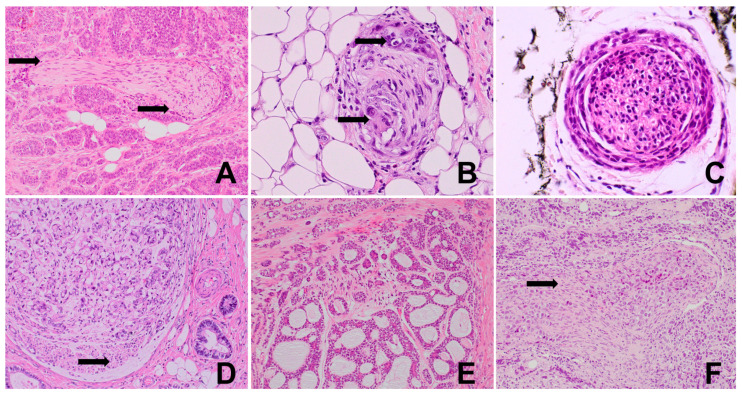
(**A**) Pancreatic neuroendocrine tumor (NET G2): neoplastic infiltrate encompassing the nerves together with small lymphocytic aggregates at the perineurium (arrows), HE, 200×. (**B**) Pancreatic adenocarcinoma: cancer cell invasion (arrow) inside the nerve accompanied by an inflammatory reaction within the adipose tissue, HE, 600×. (**C**) Melanoma infiltration within and around the subcutaneous nerve at the surgical margin (dyed in black), HE, 400×. (**D**) Gastric adenocarcinoma mixed-type poorly cohesive mucocellular component destroying thick nerve (arrow) and better-differentiated cancer glands within the fibro-adipose tissue, HE, 400×. (**E**) Salivary gland adenoid cystic carcinoma: nerve splitting and disintegration by the neoplastic infiltrate, HE, 200×. (**F**) Retroperitoneal germinoma family tumor-primitive germ cells widely infiltrate the nerve (arrow) incorporated within the tumor, HE, 400×.

**Figure 7 biomedicines-12-02335-f007:**
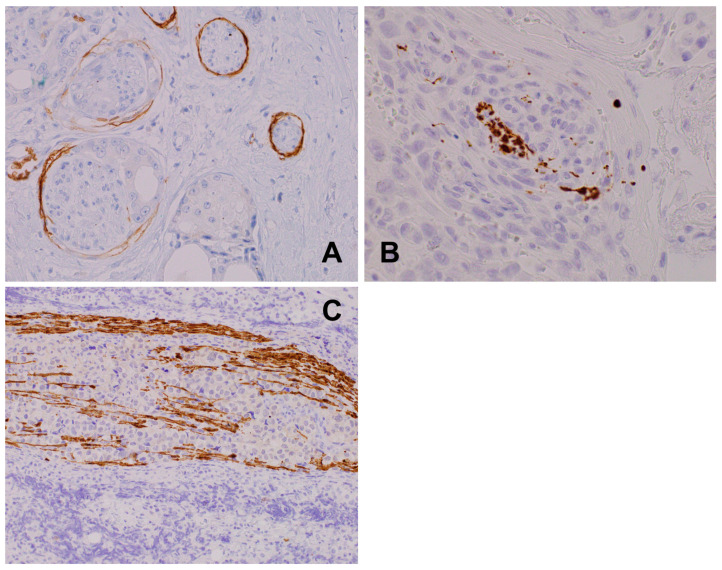
(**A**) Breast ductal adenocarcinoma cancer glands with an intense subperineurial invasion, GLUT1, 400×. (**B**) Squamous cell carcinoma of the lung: central preserved cluster of axons from the pre-existing nerve and dispersed axons inside the tumor infiltrate, PGP 9.5, 600×. (**C**) Retroperitoneal germinoma family tumor: primitive germ cells splitting and aggressively infiltrating the nerve, neurofilament protein, 400×.

**Figure 8 biomedicines-12-02335-f008:**
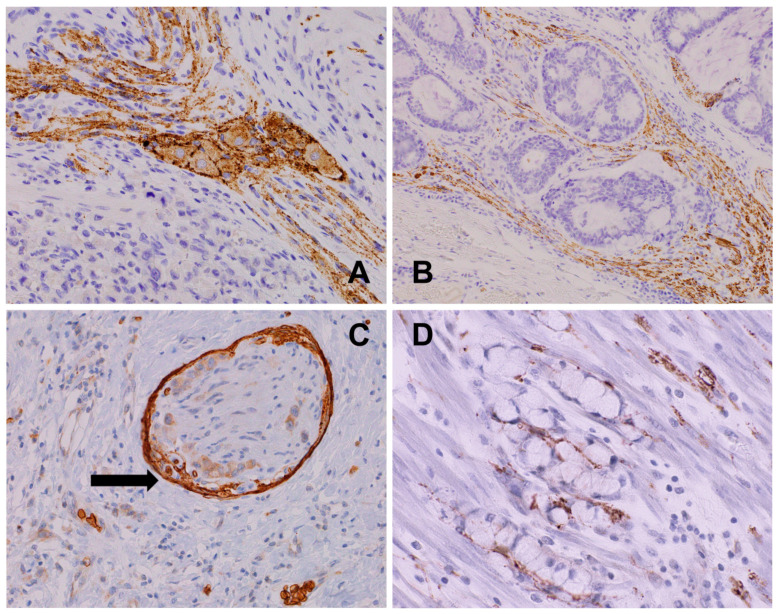
(**A**) Gastric adenocarcinoma: myenteric plexus invasion with partial separation of the fibers, synaptophysin, 400×. (**B**) Colorectal adenocarcinoma: massive intraneural invasion with destruction of the nerve branches, PGP9.5, 200×. (**C**) Gastric adenocarcinoma with scattered GLUT1 immunopositive tumor cells dispersed inside the nerve and in the subperineurial area (arrow), GLUT1, 400×. (**D**) Appendiceal goblet cell carcinoma: myenteric plexus infiltration; axons crossing tumor cell groups; PGP 9.5, 600×.

**Figure 9 biomedicines-12-02335-f009:**
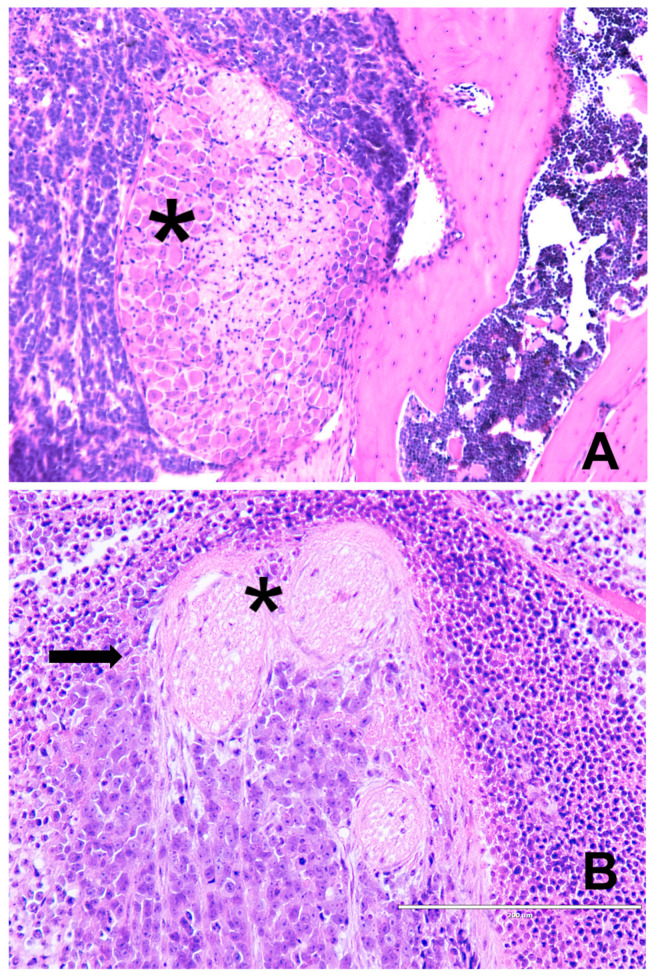
(**A**) Perineural dissemination of Ewing sarcoma (ES) cells in an experimental metastatic ES model in mice: small-round-blue-cell tumor surrounding paravertebral ganglia (*) and entering the vertebral bone, HE, 200×. (**B**) ES neoplastic cells (arrows) encompassing paravertebral ganglia (*), HE, 600×.

**Figure 10 biomedicines-12-02335-f010:**
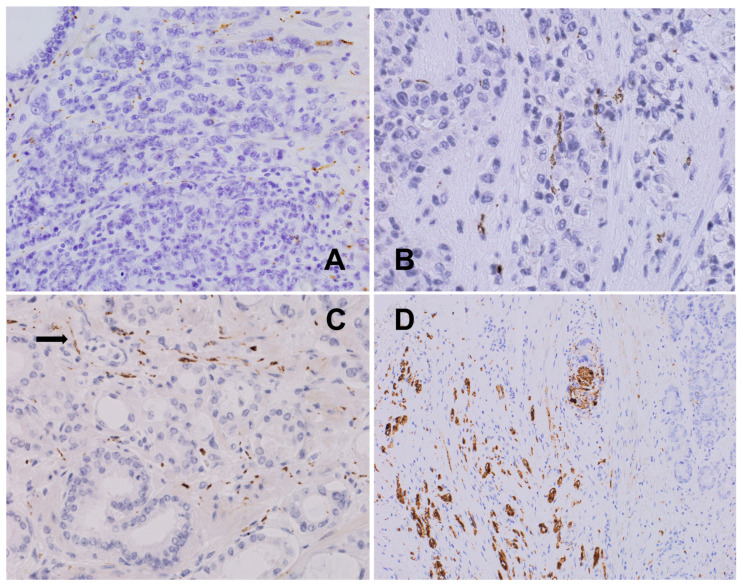
(**A**) Poorly differentiated gastric adenocarcinoma: axons dispersed among tumor cells, with heterogenous neural density, PGP 9.5, 600×. (**B**) Poorly differentiated PCa: dispersed tiny axonal structures within the cancer infiltrate, visible as dots or threads, PGP 9.5, 400×. (**C**) Cross-sections and longitudinal sections of the axons within PCa; note the pre-existing blood vessel innervation (arrow), PGP9.5, 400×. (**D**) Salivary gland adenocarcinoma NOS with high neural density, PNI, and nerve branch hypertrophy, PGP9.5, 200×.

**Figure 11 biomedicines-12-02335-f011:**
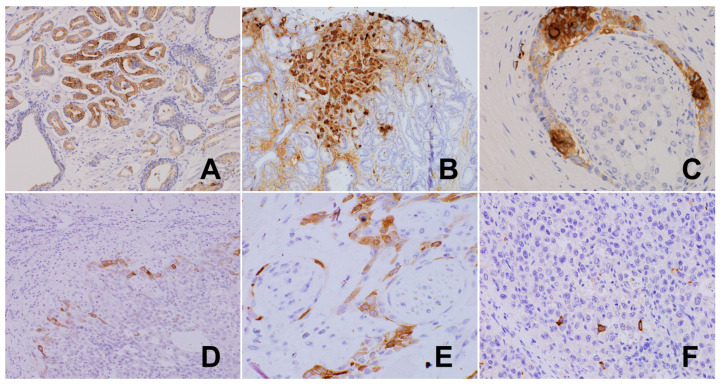
(**A**) Well-differentiated PCa-focal neuronal transdifferentiation showing tyrosine hydroxylase-positive tumor cells, TH, 200×. (**B**) PCa bone metastasis showing intense neuroendocrine differentiation, chromogranin, 200×. (**C**) PCa PNI showing groups of cells with strong chromogranin expression, whereas carcinoma within the nerve is immunonegative, chromogranin, 400×. (**D**) Squamous cell carcinoma of the base of the tongue: TH-positive tumor cells at the front of the invasion, surrounded with lympho-plasmocytic infiltrate, TH, 200×. (**E**) Squamous cell carcinoma of the base of the tongue: TH-positive tumor cells in the vicinity of encircled nerve fibers, TH, 600×. (**F**) Gastric diffuse carcinoma with single neuroendocrine cells, chromogranin, 400×.

**Figure 12 biomedicines-12-02335-f012:**
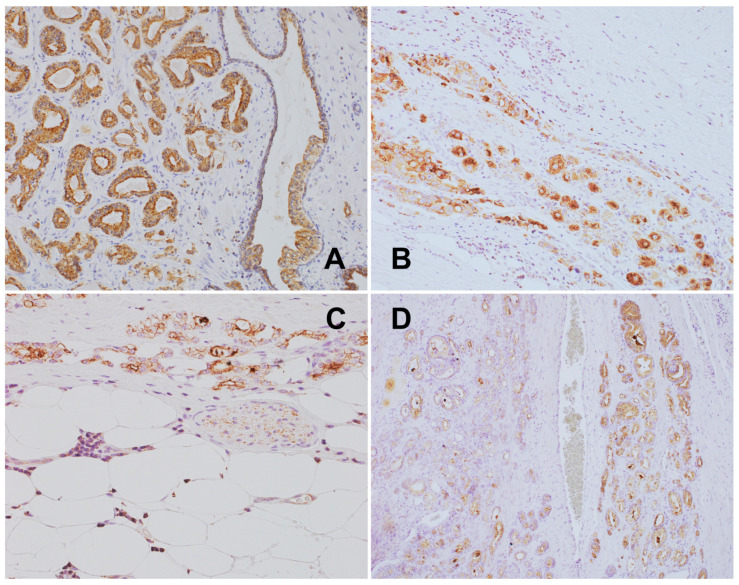
(**A**) PCa: high NPY expression in cancer glands compared to pre-existent non-neoplastic glands, NPY, 200×. (**B**) PCa-NPY-positive tumor cells with immunoreactivity equal to that of the ganglion cells, NPY, 400×. (**C**) PCa-NPY-expressing cancer cells infiltrating periprostatic fat tissue and peptide-positive intraneural axons, NPY, 400×. (**D**) PCa-NPY expression zonation with increased expression in the invasion front, NPY, 200×.

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
