# Peer review of "Remarks on Selected Morphological Aspects of Cancer Neuroscience: A Microscopic Photo Review"

_biomedicines, 2024, doi:10.3390/biomedicines12102335_

Round 1

Reviewer 1 Report

Comments and Suggestions for Authors

In the manuscript submitted for review, the Authors analyzed the available comments on selected morphological aspects of cancer neurobiology.

I believe that the topic of the manuscript is interesting and "current", and the entire work is well thought out. The Authors put a lot of work into preparing this interesting work. The reader's attention is undoubtedly drawn to the carefully prepared figures, which form the basis of this work and provide the reader with quite a lot of information.

In the introduction, the Authors provide a brief but sufficient introduction to the issue, in principle everything that is needed can be found in the introduction.

My comment only concerns the different magnifications of the photos within the same group, e.g. in Figure 1, photos A and C are magnified 200x, while photo B is 100x. Was this the intended effect?

Author Response

In our manuscript, the pictures with different magnifications were used to describe the innervation of the tumors in the most accurate manner.  The idea was to optimize the visibility of details and presented tissue / cellular elements, as well as topographic aspects in a different magnifications. The photographs were taken from our repository created over years during diagnostics and scientific projects, in consequence were taken on two types  of Olympus cameras on BX 43 Olympus microscopes using different software, with different presets. On the other hand we could remove scalebars on the selected  photos, but we could not add them to original files.

Reviewer 2 Report

Comments and Suggestions for Authors

The paper by Ewa Iżycka-Åšwieszewska, Jacek GulczyÅ„ski and co-authors represents a short review on cancer neuroscience, dominated by morphological details and microphotographs. Such large dominance of one aspect is the only significant disadvantage of the paper. Since there is no "Essay" in the journal list (https://www.mdpi.com/journal/biomedicines/instructions), the authors should choose an appropriate variant, which likely would be simply a  ("review"). Then, the significance of such a rather small and single-sided review may be questioned. However, from the scientific point of view, the paper is written well, the figures are of good quality, and the references are present where appropriate.

My remarks about the typos are listed below.

"neuroendocrine phenotype (NED)" - line 113

"neurofilament proteins (NFP H, M, L))" - line 144

"SAP 102" - line 147

"high power filed" - line 296

Author Response

The idea of our study was to present the photoreport with a short review (kind of a pictorial essay). The morphological images are not often and widely presented in the current literature. We believe that simple but detailed visualization of different cancer tissue sections explains many aspects of cancer-nerves cross-talk and can inspire the scientists for new methodological approach solutions and better understanding.

My remarks about the typos are listed below.
"neuroendocrine phenotype (NED)" - line 113
"neurofilament proteins (NFP H, M, L))" - line 144
"SAP 102" - line 147
"high power filed" - line 296

The typos were corrected.

Reviewer 3 Report

Comments and Suggestions for Authors

This review describes the morphological aspects of cancer neuroscience. Using different histological sections, the authors present different aspects of cancer invasion in nervous system and its pathologies. The manuscript needs more clarity in terms of organization of the sections:

1. what are the existing methods of diagnosis and what is the current state of the art in this area?

2. How this review advances our understanding of this area of work

3. What are current challenges and prospective solutions. 

4. Some of the figures dont even have any scale bars. None of the data is quantitative and there are mere pictures all over. The message from just the pictures is not very clear. 

Comments on the Quality of English Language

English is fine. 

Author Response

  1. what are the existing methods of diagnosis and what is the current state of the art in this area?

Cancer neuroscience is a new and important area of investigation of cancer in many aspects, the knowledge is increasing but results presented and published are  different and even contrary, because of diverse methodology of investigation. From the morphologic point of view there are several fields for research: PNI, nerve/ axonal density, nerve network topography and neuroendocrine differentiation.

Perineural invasion (PNI) is the most common form of cancer and nerve interaction, assessed in histopathological examination. Currently, the nerve/ axonal density is assessed only in preclinical studies. The PNI is determined as an obligatory element  in oncological pathological reports, and it is a prognostic factor. Traditional histopathological examination remains the primary method for diagnosing PNI, however the definition and assessment method are not clearly described. Usually tissue samples from biopsies or surgical resections are stained and examined to identify cancer cells around nerves, however the routine H&E staining allows visualisation of bigger nerve branches. Specific neural markers like S100 protein or neurofilament protein can be used to highlight nerves and detect PNI more precisely. Many more immunohistochemical nerve markers are used in research. The current state of art in our paper was described in ,, Basic methods and tissue markers’’chapter (130-161).

In the literature, nerve/ axon density values have wide range, eg. in prostate cancer it is reported quantitatively  from less than 5 to about 100 per field, or just as “low” vs “high” depending on methodological approach. (The paragraph added to the original text).

  1. How this review advances our understanding of this area of work

The biggest advances of our work is the presentation of various forms of cancer and nerves interaction which is lacking in  literature. Our work constitutes the good background for researchers who work in the area of cancer neuroscience.

  1. What are current challenges and prospective solutions.

The current challenges in cancer neuroscience in relation to morphological aspects includes the understanding the complex, bidirectional communication network between cancer and nerves which contributes to cancer progression, metastasis and resistance to therapy. The future challenges in cancer neuroscience include advanced imaging techniques, molecular profiling, AI-driven digital pathology, understanding the molecular drivers, specific biomarkers and translation from the lab to the clinical practice which may lead to new therapeutic strategies (lines 127-128, 216-218, 258-277, 325-329)

  1. Some of the figures don’t even have any scale bars. None of the data is quantitative and there are mere pictures all over. The message from just the pictures is not very clear.

The pictures were corrected and unified. Scalebars were removed and photos adjusted, we could not add artificial scalebars. In consequence we utilize only magnification values in the text. The idea of our study was to present the photoreport with a short review (a kind of a pictorial essay). The morphological images are not often and widely presented in the modern publications, we believe that simple visualization of cancer neuroscience explains many aspects of cancer- nerves cross-talk and inspires for some methodological solutions in scientific work. The images, as we mentioned above,  come from our repository collected over several years.

Additionally we removed one reference, because the paper was retracted.

  1. Kamiya A, Hayama Y, Kato S, et al. Genetic manipulation of autonomic nerve fiber innervation and activity and its effect on breast cancer progression. Nat Neurosci. 2019;22(8):1289-1305. doi:10.1038/s41593-019-0430-3

Round 2

Reviewer 3 Report

Comments and Suggestions for Authors

the revised manuscript is now recommended for publication